# Whole genomes from the extinct Xerces Blue butterfly can help identify declining insect species

Toni de-Dios[1,2†], Claudia Fontsere[1,3†], Pere Renom[1†], Josefin Stiller[4], Laia Llovera[1], Marcela Uliano-Silva[5], Alejandro Sánchez-Gracia[6], Charlotte Wright[5], Esther Lizano[1,7], Berta Caballero[8], Arcadi Navarro[1,9], Sergi Civit[6], Robert K Robbins[10], Mark Blaxter[5], Tomàs Marquès[1,7,9,11]*, Roger Vila[1]*, Carles Lalueza-Fox[1,8]*

[1]Institute of Evolutionary Biology, Barcelona, Spain; [2]Institute of Genomics, University of Tartu, Tartu, Estonia; [3]Section for Evolutionary Genomics, The Globe Institute, Faculty of Health and Medical Sciences, University of Copenhagen, Copenhagen, Denmark; [4]Centre for Biodiversity Genomics, University of Copenhagen, Copenhagen, Denmark; [5]Wellcome Sanger Institute, Saffron Walden, United Kingdom; [6]Departament of Genetics, Microbiology and Statistics-Institut de Recerca de la Biodiversitat (IRBio), Universitat de Barcelona, Barcelona, Spain; [7]Institut Català de Paleontologia Miquel Crusafont, Universitat Autònoma de Barcelona, Barcelona, Spain; [8]Museu de Ciències Naturals de Barcelona, Barcelona, Spain; [9]Catalan Institution of Research and Advanced Studies (ICREA), Barcelona, Spain; [10]Department of Entomology, National Museum of Natural History, Smithsonian Institution, Washington, United States; [11]CNAG-CRG, Centre for Genomic Regulation, Barcelona Institute of Science and Technology (BIST), Barcelona, Spain

*For correspondence:
tomas.marques@upf.edu (TM);
roger.vila@csic.es (RV);
carles.lalueza.fox@gmail.com (CL-F)

†These authors contributed equally to this work

**Abstract** The Xerces Blue (*Glaucopsyche xerces*) is considered to be the first butterfly to become extinct in historical times. It was notable for its chalky lavender wings with conspicuous white spots on the ventral wings. The last individuals were collected in their restricted habitat, in the dunes near the Presidio military base in San Francisco, in 1941. We sequenced the genomes of four 80- to 100-year-old Xerces Blue, and seven historical and one modern specimens of its closest relative, the Silvery Blue (*Glaucopsyche lygdamus*). We compared these to a novel annotated genome of the Green-Underside Blue (*Glaucopsyche alexis*). Phylogenetic relationships inferred from complete mitochondrial genomes indicate that Xerces Blue was a distinct species that diverged from the Silvery Blue lineage at least 850,000 years ago. Using nuclear genomes, both species experienced population growth during the Eemian interglacial period, but the Xerces Blue decreased to a very low effective population size subsequently, a trend opposite to that observed in the Silvery Blue. Runs of homozygosity and deleterious load in the former were significantly greater than in the later, suggesting a higher incidence of inbreeding. These signals of population decline observed in Xerces Blue could be used to identify and monitor other insects threatened by human activities, whose extinction patterns are still not well known.

## eLife assessment

This **important** study illustrates the value of museum samples for understanding past genetic variability in the genomes of populations and species, including those that no longer exist. The authors present genomic sequencing data for the extinct Xerces Blue butterfly and report **convincing**

evidence of declining population sizes and increases in inbreeding beginning 75,000 years ago, which strongly contrasts to the patterns observed in similar data from its closest relative, the extant Silvery Blue butterfly. Such long-term population health indicators may be used to highlight still extant but especially vulnerable-to-extinction insect species – irrespective of their current census population size abundance.

## Introduction

The Xerces Blue butterfly (*Glaucopsyche xerces*) (*Boisduval, 1852*) was native to the coastal sand dunes of San Francisco in association with the common Deerwood (*Acmispon glaber*), which was the preferred food source for larval stage (*Tilden, 1956*). It was notable for its iridescent blue colouration on the dorsal (upper) wing surface, and conspicuous, variable white spots on the ventral surface (*Downey, 1956*). With the growth of San Francisco and the destruction of sand dune habitats, the Xerces Blue became restricted to a few sites in what is now Golden Gate National Recreation Area. The last specimens were reportedly collected by entomologist W. Harry Lange on 23 March 1941 (*Downey, 1956*). It is considered the first butterfly to have been driven to global extinction by human activities (*Downey, 1956*).

The Xerces Blue and the closely related Silvery Blue (*Glaucopsyche lygdamus*) were recently proposed to be distinct species based on mtDNA data from a single Xerces Blue specimen (*Grewe et al., 2021*). However, two nuclear genes analysed (ribosomal 28S and histone H3) were invariable and genome-wide data were unavailable for the Xerces Blue, hampered by the inherent difficulties of retrieving genome-wide data from historical insect specimens (*Thomsen et al., 2009*; *Staats et al., 2013*) and the absence of a suitable reference genome. The genus *Glaucopsyche* consists of 18 extant species distributed across the temperate regions of the northern hemisphere. To provide a relevant reference, we generated an annotated genome from the Palearctic Green-Underside Blue butterfly *Glaucopsyche alexis* (*Hinojosa Galisteo et al., 2021*). Using DNA extracted from five Xerces Blue and seven Silvery Blue (*G. lygdamus*) historical specimens from the vicinity of San Francisco, and also from a modern Silvery Blue male from Canada, we generated whole-genome resequencing data for both species and investigated their relationships and historical population genetics.

## Results
### Historic and modern butterfly genomes

We extracted DNA from 12 historical specimens (5 *G. xerces*, 7 *G. lygdamus*) (*Table 1*). One Xerces Blue sample did not yield detectable DNA in two independent extractions. For each of the successful

**Table 1.** List of historical specimens analysed in this study.

| Genome # | Species | Subspp. | State | Locality | Date | Collection |
|---|---|---|---|---|---|---|
| USNMENT101413 | *G. xerces* | | California | San Francisco | NA | Barnes |
| USNMENT101402 | *G. xerces* | | California | San Francisco | 16/4/1923 | Barnes |
| USNMENT101441 | *G. xerces* | | California | San Francisco | NA | Barnes |
| USNMENT101406 | *G. xerces* | | California | San Francisco | NA | Barnes |
| USNMENT101434 | *G. xerces* | | California | San Francisco | 16/4/1923 | Barnes |
| USNMENT00181297 | *G. lygdamus* | *incognitus* | California | Marin Country | NA | Barnes |
| USNMENT00181298 | *G. lygdamus* | *incognitus* | California | Fairfax | 27/5/1932 | WMD Field |
| USNMENT00181299 | *G. lygdamus* | *incognitus* | California | Oakland | 14/4/1948 | Graham Heid |
| USNMENT00181300 | *G. lygdamus* | *incognitus* | California | San Jose | 27/3/1964 | Opler |
| USNMENT00181301 | *G. lygdamus* | *incognitus* | California | Haywood City | 1/5/1931 | WMD Field |
| USNMENT00181302 | *G. lygdamus* | *incognitus* | California | Santa Cruz | 1/4/1932 | JW Tilden/Field |
| USNMENT00181303 | *G. lygdamus* | *incognitus* | California | Santa Cruz | 8/4/1927 | GW Rawson |

**Table 2.** Mapping statistics of the analysed historical specimens.
Mapping statistics of the four historical *G. xerces* (L003, L005, L007, and L009) and the seven historical *G. lydagmus* (L002, L004, L006, L008, L011, L012, and L013) specimens mapped against the *G. alexis* reference genome. Average depth is displayed for the covered regions of each individual.

| Sample identifier | Generated reads | Q25 unique mapped reads | Breadth of coverage (%) | Average depth covered regions |
|---|---|---|---|---|
| L002 | 300,294,248 | 23,337,751 | 37.27 | 5.105 |
| L003 | 405,198,060 | 32,547,820 | 36.86 | 6.78 |
| L004 | 357,165,438 | 28,722,185 | 38.77 | 6.55 |
| L005 | 776,312,378 | 56,459,037 | 45.7 | 12.42 |
| L006 | 359,520,168 | 28,498,720 | 40.07 | 6.18 |
| L007 | 348,916,870 | 26,758,356 | 34.79 | 6.21 |
| L008 | 508,120,156 | 32,107,192 | 42.08 | 7.422 |
| L009 | 322,955,384 | 39,312,617 | 40.6 | 8.02 |
| L011 | 236,886,534 | 24,165,282 | 38.6 | 5.40 |
| L012 | 328,359,669 | 18,683,738 | 33.37 | 4.29 |
| L013 | 385,635,644 | 52,612,937 | 47.2 | 12.3 |

extracts, we prepared a single library which was shotgun sequenced on the HiseqX Illumina platform. We mapped 124,101,622 and 184,084,237 unique DNA reads of Xerces Blue and Silvery Blue, respectively, against the *G. alexis* reference genome (*Table 2*). The DNA reads exhibited typical ancient DNA features, such as short mean read length (ranging from 47.55 to 67.41 bases on average), depending on the specimen and post-mortem deamination patterns at the 5′ and 3′ ends (*Supplementary file 1A*). As listed in the original museum records, we found one Silvery Blue and two Xerces Blue females. Inter-individual comparisons suggested no close kinship link among the studied individuals.

The historical genomes covered 49.3% (Xerces Blue) and 55.2% (Silvery Blue) of the *G. alexis* reference genome, largely because repetitive chromosomal regions cannot be confidently assessed with short, ancient DNA sequence reads (*Supplementary file 1B*). To estimate the mappable fraction of

**Table 3.** Coordinates of the analysed colouration genes.
Genomic coordinates in *G. alexis* reference genomes of different wing colouration genes described in other butterfly species.

| Chromosome | Start | End | Gene |
|---|---|---|---|
| FR990043.1 | 5,387,706 | 5,403,599 | Wnt1 |
| FR990043.1 | 5,417,902 | 5,423,677 | Wnt6 |
| FR990043.1 | 5,519,353 | 5,539,737 | Wnt10b |
| FR990043.1 | 5,553,666 | 5,554,753 | Wnt10a |
| FR990043.1 | 26,972,856 | 26,974,487 | WntA |
| FR990046.1 | 2,343,467 | 2,357,667 | Wnt7b |
| FR990046.1 | 6,255,275 | 6,271,623 | Wnt5b |
| FR990046.1 | 19,475,636 | 19,486,554 | Wnt9 |
| FR990050.1 | 16,200,978 | 16,212,495 | Wnt11 |
| FR990054.1 | 20,633,400 | 20,655,261 | Cortex |
| FR990059.1 | 20,254,460 | 20,255,275 | Optix |

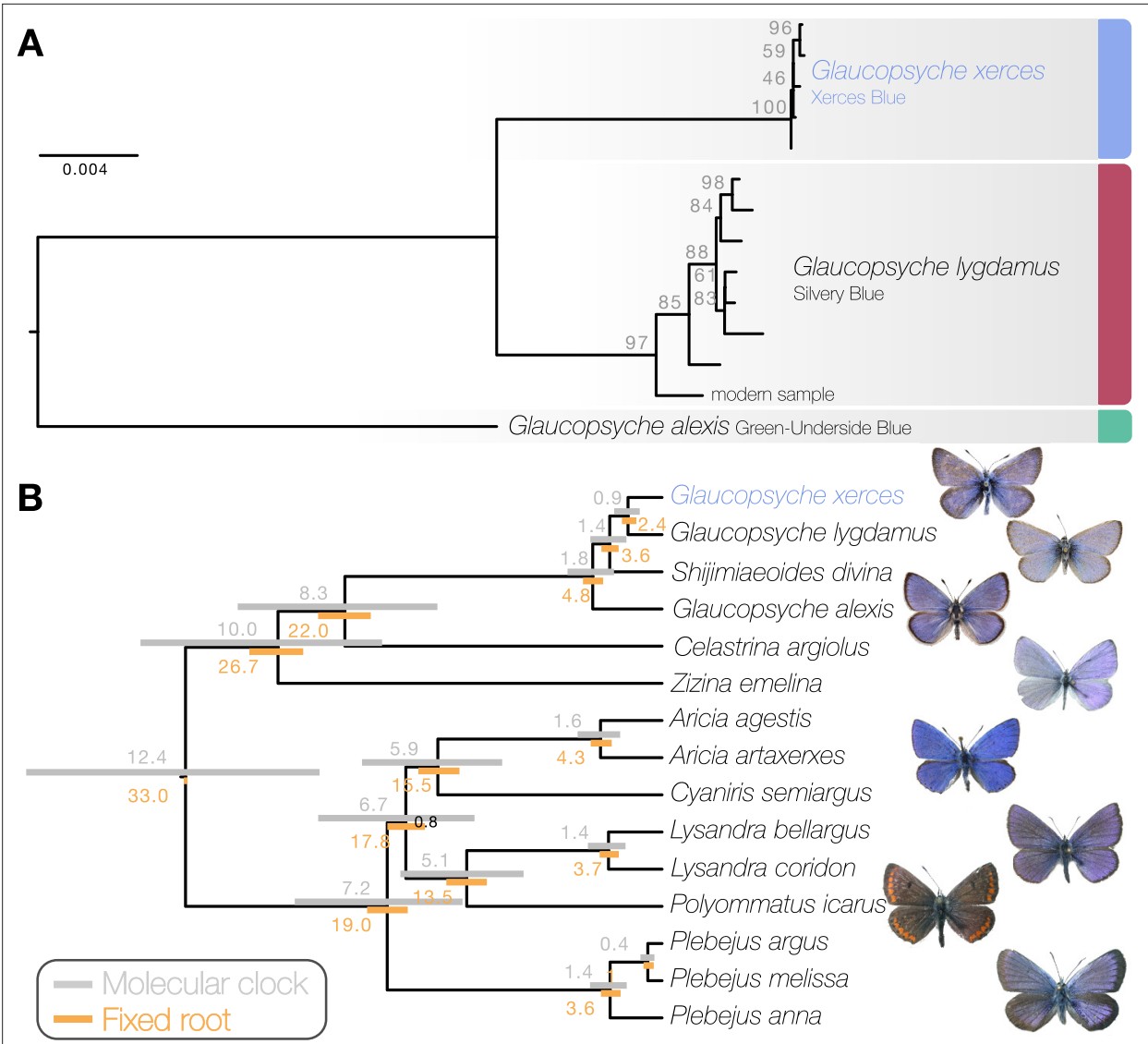

**Figure 1.** Phylogenetic placement of the Xerces Blue. (**a**) Maximum likelihood tree from whole mitochondrial genomes of Xerces Blue, Silvery Blue, and Green-Underside Blue. Node labels are bootstrap support values. (**b**) Time-calibrated phylogeny from Bayesian inference using mitochondrial protein-coding genes of Xerces Blue and related butterflies. Node values show median age estimates from dating analysis with a molecular clock (above nodes) or from fixing the age of the root (below nodes). Bars are 95% HPD intervals for node ages. All posterior probabilities were 1, except for one node annotated in black.

the reference *G. alexis* genome, we randomly fragmented it to 50–70 nucleotides and mapped the generated fragments back to the complete genome. An average of 57.8% of the *G. alexis* genome was covered with these read lengths. We suggest that reduced coverage from the historical specimens may be due to genomic divergence of *G. xerces* and *G. lygdamus* from the *G. alexis* reference. The annotation of genes located in those unrecoverable regions provided a putative list of 14 nuclear genes with diverse functions obtained from BLAST, that should be further explored to understand the uniqueness of the extinct species (*Table 3*).

## Phylogenetic relationships

Maximum likelihood phylogenetic inference using whole mitochondrial genomes showed that the Xerces Blue specimens form a monophyletic clade, as do the Silvery Blue specimens (*Figure 1a*). We inferred a time-calibrated Bayesian phylogenetic tree from protein-coding genes analysis and 12 related butterflies in Polyommatinae subfamily (*Supplementary file 1C*), revealing high support for

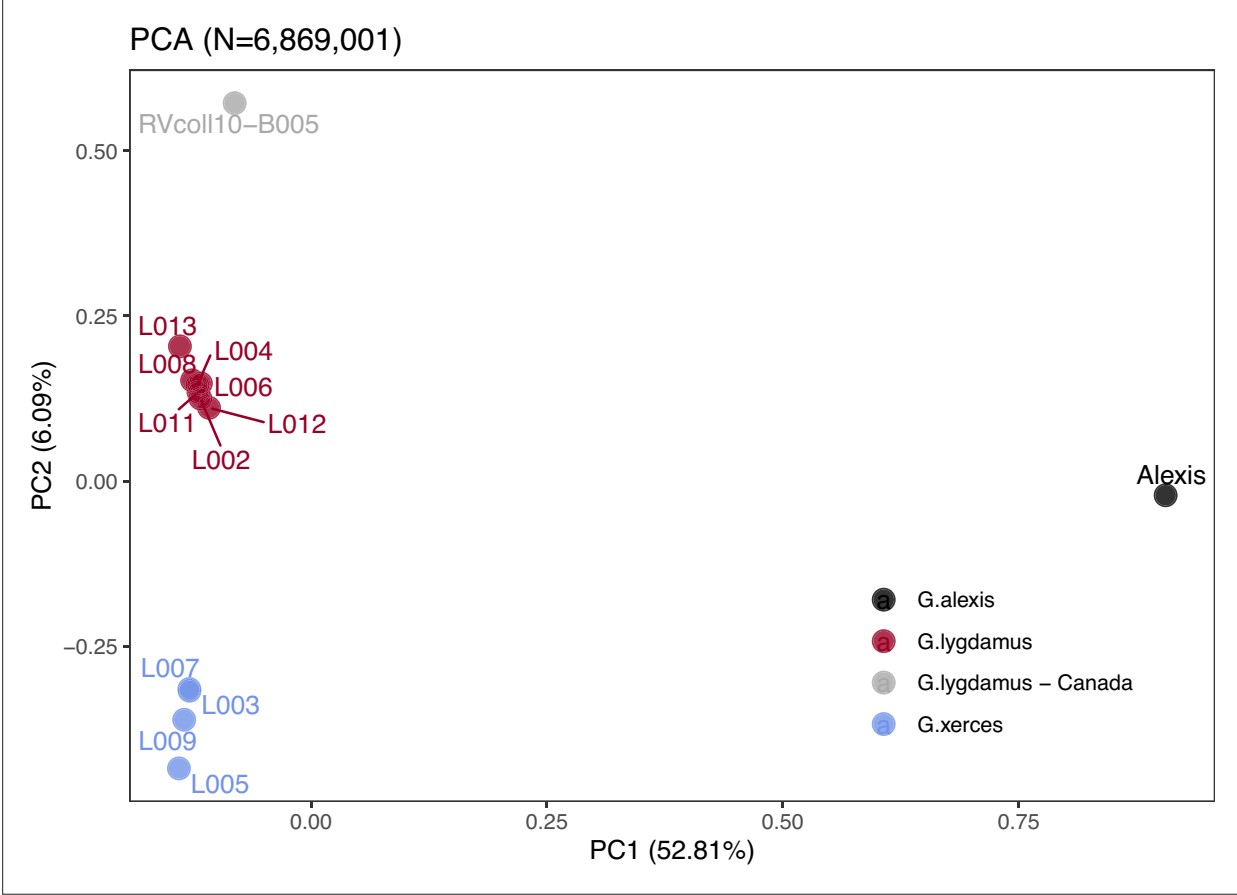

**Figure 2.** Plotting of PC1 and PC2 of the principal component analysis (PCA). The PCA was generated with nuclear DNA data (*N* = 6,682,591 SNPs (single nucleotide polymorphisms)) from 11 historical butterfly specimens (4 *G. xerces* and 7 *G. lygdamus*), a modern *G. lygdamus* from Canada (RVcoll10-B005) and a modern *G. alexis* reference genome. The PCA shows a clear separation of both historical species and the reference in the first PC (explaining 52.81% of the variance), and separation of *G. xerces* and *G. lygdamus* by the second PC (explaining 6.09% of the variance), supporting they are separated lineages.

the sister group relationship (posterior probability = 1). We found the specie *Shijimiaeoides (Sinia) divina* inside the *Glaucopsyche* clade, in agreement with previous phylogenetic studies (*Lukhtanov and Gagarina, 2022*). Because there are no known fossils to calibrate the time since divergence, we first used a molecular clock that spanned the range of rates frequently used for arthropod mitochondrial genes (1.5–2.3% divergence/Ma). Our dated analysis yielded an origin of this subgroup of Polyommatinae at 12.4 Ma (8.82–16.27 Ma 95% HPD [highest posterior density] interval) and divergence of the Xerces Blue from the Silvery Blue at 900,000 years ago (0.61–1.19 Ma 95% HPD interval, *Figure 1b*). A second estimate based on larger-scale fossil-based calibrations (*Espeland et al., 2018*) fixed the origin of the subgroup to ca. 33 Ma (*Chazot et al., 2019*), inferred the subsequent divergence of the Xerces Blue and Silvery Blue to 2.40 Ma (1.95–2.73 Ma 95% HPD interval, *Figure 1b*). The recent speciation of Xerces and Silvery Blue is not obviously due to infection with the *Wolbachia*, as no evidence of infection of the sampled specimens with this alpha-proteobacterium is detected in the raw read data.

Principal component analysis (PCA) using PCAngsd (*Meisner and Albrechtsen, 2018*) and nuclear genome polymorphisms for the three *Glaucopsyche* species supports the relationships among them; the historical specimens are equally distant to the Green-Underside Blue (*G. alexis*) in the first principal components (PC), explaining 52.81% of the variance (*Figure 2*). The second PC separates the Xerces Blue from the Silvery Blue specimens.

## Demographic history and diversity

We used the pairwise sequentially Markovian coalescent (PSMC) algorithm (*Li and Durbin, 2011*) to evaluate the demographic histories of both butterfly species, first exploring the two specimens with

highest coverage (L05 and L13) (*Figure 3*). We found an increase in effective population size in both species that is roughly coincident with the interglacial Marine Isotopic Stage 7 (approximately from 240,000 to 190,000 years ago; *Batchelor et al., 2019*). After this timepoint the trends differ. We estimated a continuous decrease in Xerces Blue population size in parallel to the Wisconsin Glacial Episode, which started about 75,000 years ago. However, both the modern and the historical Silvery Blue do not appear to have been negatively affected by this event (*Figure 3—figure supplement 1*), suggesting different adaptive strategies to cope with cooling temperatures and/or food plant availability.

Second, we generated PSMC curves from the remaining lower-coverage individuals and down-sampled data from specimen L05 to 50% and 75% of the total coverage to explore the effects of coverage on estimation of heterozygous sites. Although there was a reduction in the effective population size estimates, as expected, the temporal trajectories in lower-coverage individuals were similar to their respective, higher-coverage Xerces Blue and Silvery Blue references (*Figure 3—figure supplement 1*).

We subsequently explored the heterozygosity of each individual and found that Xerces Blue had 22% less heterozygosity on average than the Silvery Blue historical samples, a difference that is statistically significant (*T*-test; p = 0.0072) (*Figure 4*, *Supplementary file 1D*). We searched for runs of homozygosity (RoH) that can indicate the existence of inbreeding in a dwindling population. The total fraction of the genome presenting RoH, although limited, is much higher in Xerces Blue (up to 6% of the genome) than in Silvery Blue, especially in short RoH of size between 100 and 500 kb (*Figure 4—figure supplement 1*), consistent with background inbreeding. The limited presence of long RoH discards consanguinity as a common scenario in Xerces Blue.

We identified amino acid-changing alleles that may be suggestive of a deleterious genetic load associated with long-term low population numbers in the Xerces Blue. The average Ka/Ks ratio is higher in Xerces Blue than in Silvery Blue; the former also carries a higher fraction of nonsense and functionally high-to-moderate effect variants in homozygosity and RoH with an increased concentration of high-to-moderate effect variants (*Figure 5*), as predicted with a functional prediction toolbox, SnpEff (*Cingolani et al., 2012*).

## Discussion

We have used a modern reference genome and ancient DNA genome sequence data from museum specimens to explore the relationships and historical population genetic history of an extinct butterfly, the Xerces Blue; to our knowledge, this is the first ancient genome ever generated from an extinct insect. Based upon a near-complete mtDNA genome from a Xerces Blue specimen (*Grewe et al., 2021*) proposed that the Xerces Blue and the Silvery Blue were distinct species. We confirm this finding using full mitochondrial genomes and extensive nuclear genomic data from multiple specimens. Given the lack of evidence for *Wolbachia* infection, the recent speciation of Xerces Blue and Silvery Blue seems unrelated to cytoplasmic incompatibility cause by this endosymbiont (*Telschow et al., 2005*; *Sucháčková Bartoňová et al., 2021*); a detailed analysis of genomic architectures could help identify barriers to introgression between these species.

Our analyses indicate that the Xerces Blue had experienced a severe demographic decline for tens of thousands of years, likely associated with changing climatic factors. Thus, the destruction of the Xerces Blue habitat by humans was likely the final blow in the extinction process. We provide evidence for low population size in Xerces Blue, correlated with low genetic variation, a higher proportion of RoH and increased frequency of deleterious, amino acid-changing alleles (*Szpiech et al., 2013*; *Spielman et al., 2004*; *Palkopoulou et al., 2015*). However, there was no genetic evidence of recent inbreeding.

Inbreeding genetic signals in the form of long chromosomal sections with no variation sometimes occur in critically endangered species (*van der Valk et al., 2019*; *Díez-Del-Molino et al., 2018*) and in extinct species such as the last Mammoths from Wrangel Island (*Rogers and Slatkin, 2017*) or the Altai Neanderthal (*Prüfer et al., 2014*). The PSMC shows a continuous low effective population size for Xerces Blue; demographic declines are also seen in some extinct species, including Wrangel Mammoths (*Palkopoulou et al., 2015*) but not in others such as the Woolly Rhino that showed a pre-extinction demographic stability and relatively low inbreeding signals (*Lord et al., 2020*). In many endangered species there is little concordance between genome diversity, population sizes,

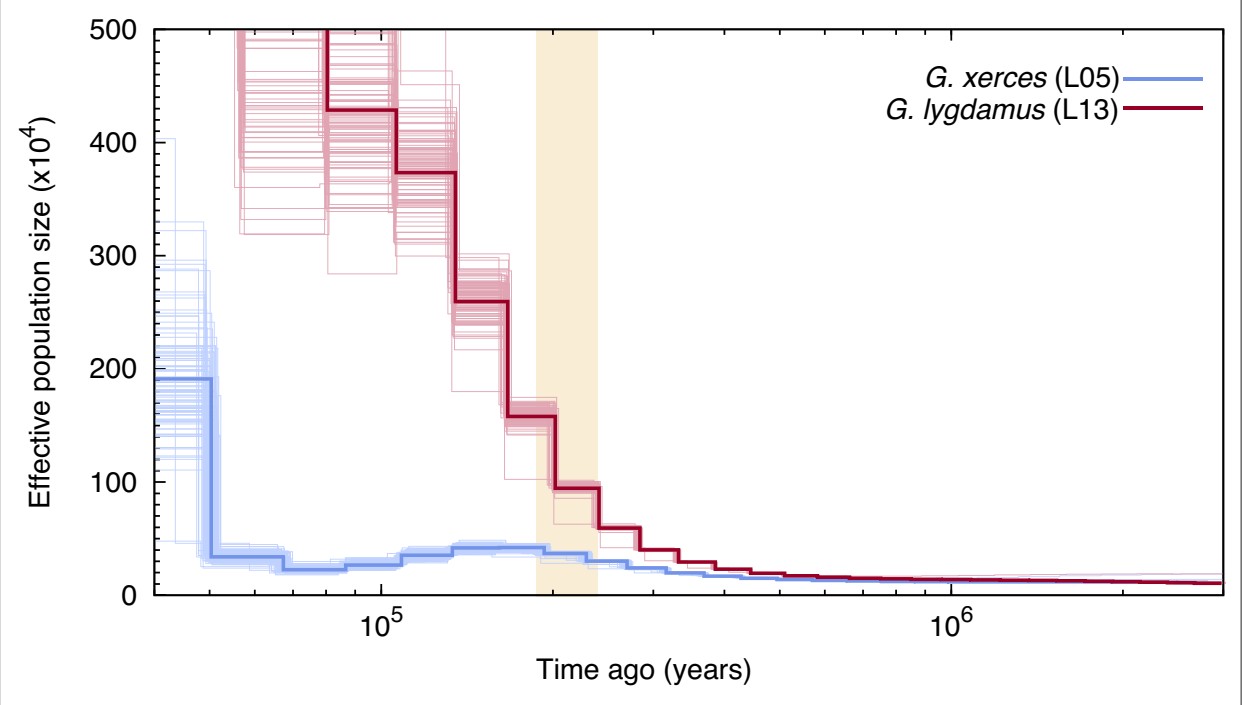

**Figure 3.** Pairwise sequentially Markovian coalescent (PSMC) plot of one Xerces Blue (*Glaucopsyche xerces*) (L05) specimen and one Silvery Blue specimen (*Glaucopsyche lygdamus*). The two historical samples are those with higher average coverage. Individual PSMC plots were bootstrapped 100 times each (lighter lines). One year of generation time and a mutation rate of $\mu = 1.9 \times 10^{-9}$ were used. The peak of the Marine Isotopic Stage 7 interglacial is marked in yellow.

The online version of this article includes the following figure supplement(s) for figure 3:

**Figure supplement 1.** Pairwise sequentially Makovian coalescent (PSMC) plots of Xerces Blue and Silvery Blue.

and conservation status (***Díez-Del-Molino et al., 2018***); this decoupling was also observed in the genomes of the extinct passenger pigeon that despite being one of the world's most numerous vertebrates, showed a surprisingly low genetic diversity (***Murray et al., 2017***). Despite being notoriously abundant, insects, and in particular butterflies, are very sensitive to climate fluctuations; therefore, we suggest that insects with observations of demographic traits indicative of long-term low effective population size such as those found in Xerces Blue should be considered to be especially vulnerable to extinction events.

Our study further demonstrates the value of museum insect specimens for estimating temporal changes in genetic diversity at a population scale (***Lalueza-Fox, 2022***). Despite being notoriously abundant, insects, and in particular butterflies, are very sensitive to climate fluctuations. We suggest that insects with genetic observations of long-term low effective population size such as those found in Xerces Blue should be considered to be especially vulnerable to extinction events. However, being the insect numbers usually very high, it is likely that their genomic signals of extinction could be different to those described in vertebrates in many cases. Therefore, this is a subject that should be further explored with genomic data from other declining insects.

## Methods

**Key resources table**

| Reagent type (species) or resource | Designation | Source or reference | Identifiers | Additional information |
|---|---|---|---|---|
| Biological sample (*Glaucopsyche xerces*; female) | L003 | This paper | SAMEA114094142 | See Materials and methods |

*Continued on next page*

*Continued*

| Reagent type (species) or resource | Designation | Source or reference | Identifiers | Additional information |
|---|---|---|---|---|
| Biological sample (*G. xerces*; male) | L005 | This paper | SAMEA114094143 | See Materials and methods |
| Biological sample (*G. xerces*; male) | L007 | This paper | SAMEA114094144 | See Materials and methods |
| Biological sample (*G. xerces*; female) | L009 | This paper | SAMEA114094145 | See Materials and methods |
| Biological sample (*Glaucopsyche lygdamus*; male) | L002 | This paper | SAMEA114094134 | See Materials and methods |
| Biological sample (*G. lygdamus*; male) | L004 | This paper | SAMEA114094135 | See Materials and methods |
| Biological sample (*G. lygdamus*; male) | L006 | This paper | SAMEA114094136 | See Materials and methods |
| Biological sample (*G. lygdamus*; male) | L008 | This paper | SAMEA114094137 | See Materials and methods |
| Biological sample (*G. lygdamus*; male) | L011 | This paper | SAMEA114094138 | See Materials and methods |
| Biological sample (*G. lygdamus*; female) | L012 | This paper | SAMEA114094139 | See Materials and methods |
| Biological sample (*G. lygdamus*; male) | L013 | This paper | SAMEA114094140 | See Materials and methods |
| Biological sample (*G. lygdamus*; male) | RVcoll10-B005 | This paper | SAMEA114094141 | See Materials and methods |
| Biological sample (*Glaucopsyche alexis*; male) | G. alexis | **Hinojosa Galisteo et al., 2021** | ilGlaAlex1.1; GCA_905404095.1 | |
| Biological sample (*Aricia agestis*) | A. agestis | **Hayward et al., 2023** | LR990279.1 | |
| Biological sample (*Aricia artaxerxes*) | A. artaxerxes | **Ebdon et al., 2022** | OW569311.1 | |
| Biological sample (*Celastrina argiolus*) | C. argiolus | **Hayward et al., 2021** | LR994603.1 | |
| Biological sample (*Cyaniris semiargus*; male) | C. semiargus | **Lohse et al., 2023** | LR994570.1 | |
| Biological sample (*G. alexis*; male) | G. alexis | **Hinojosa Galisteo et al., 2021** | FR990065.1 | |
| Biological sample (*G. xerces*) | G. xerces | **Grewe et al., 2021** | MW677564.1 | |
| Biological sample (*Lysandra bellargus*; female) | L. bellargus | **Lohse et al., 2022** | HG995365.1 | |
| Biological sample (*Lysandra coridon*; male) | L. coridon | **Vila et al., 2023** | HG992145.1 | |
| Biological sample (*Plebejus argus*) | P. argus | **Zhou et al., 2020** | MN974526.1 | |
| Biological sample (*Plebejus melissa*) | P. melissa | **Ellis et al., 2021** | DWQ001000057.1 | |
| Biological sample (*Plebejus anna*) | P. anna | **Ellis et al., 2021** | DWTA01000073.1 | |
| Biological sample (*Polyommatus icarus*; male) | P. icarus | https://www.darwintreeoflife.org/ | OW569343.1 | |
| Biological sample (*Shijimiaeoides divina*) | S. divina | **Jeong et al., 2017** | NC_029763.1 | |

*Continued on next page*

*Continued*

| Reagent type (species) or resource | Designation | Source or reference | Identifiers | Additional information |
|---|---|---|---|---|
| Biological sample (*Zizina emelina*) | *Z. emelina* | *Liu et al., 2020* | MN013031.1 | |
| Software, algorithm | BUSCO | *Manni et al., 2021* | | v.5.1.2 |
| Software, algorithm | AdapterRemoval | *Schubert et al., 2016* | | v.2.2.2 |
| Software, algorithm | BWA – backtrack | *Li and Durbin, 2009* | | v.0.7.1 |
| Software, algorithm | BWA – mem | *Li, 2013* | | v.0.7.1 |
| Software, algorithm | Qualimap2 | *Okonechnikov et al., 2016* | | v.2.2.2 |
| Software, algorithm | pmdtools | *Skoglund et al., 2014* | | v.0.50 |
| Software, algorithm | MapDamage2 | *Jónsson et al., 2013* | | v.2.7.12 |
| Software, algorithm | Bedtools | *Quinlan and Hall, 2010* | | v.2.27.1 |
| Software, algorithm | snpAD | *Prüfer, 2018* | | v.0.3.2 |
| Software, algorithm | GATK | *McKenna et al., 2010* | | v.3.5–3.7 |
| Software, algorithm | vcftools | *Danecek et al., 2011* | | v.0.1.12b–0.1.14b |
| Software, algorithm | angsd | *Korneliussen et al., 2014* | | v.0.916 |
| Software, algorithm | bcftools | *Danecek et al., 2021* | | v.1.9 |
| Software, algorithm | Mitofinder | *Allio et al., 2020* | | v.1.4 |
| Software, algorithm | MACSE | *Ranwez et al., 2018* | | v.2.05 |
| Software, algorithm | MAFFT | *Katoh and Standley, 2013* | | v.7.490 |
| Software, algorithm | IQ-TREE2 | *Minh et al., 2020* | | v.2.1.3 |
| Software, algorithm | ModelFinder | *Kalyaanamoorthy et al., 2017* | | Available in IQ-TREE2 |
| Software, algorithm | UFBoot2 | *Hoang et al., 2018* | | Available in IQ-TREE2 |
| Software, algorithm | BEAST2 | *Bouckaert et al., 2019* | | v.2.6.3 |
| Software, algorithm | bModelTest | *Bouckaert and Drummond, 2017* | | v.1.2.1 |
| Software, algorithm | Tracer | *Rambaut et al., 2018* | | v.1.7.2 |
| Software, algorithm | PSMC | *Li and Durbin, 2011* | | v.0.6.5 |
| Software, algorithm | PCAngsd | *Meisner and Albrechtsen, 2018* | | v.20180209 |
| Software, algorithm | Bcftools-roh | *Narasimhan et al., 2016* | | v.1.9 |
| Software, algorithm | SNPeff | *Cingolani et al., 2012* | | v.4.3 |
| Software, algorithm | Picard | *Broad Institute, 2015* | | v.2.0.1 |
| Software, algorithm | Samtools | *Li et al., 2009* | | v.1.6 |
| Software, algorithm | BamUtil | *Jun et al., 2015* | | v.1.0.13 |
| Software, algorithm | Bedtools | *Quinlan and Hall, 2010* | | v.2.27.1 |
| Software, algorithm | BLAST | *Altschul et al., 1990* | | v.2.2.2 |
| Software, algorithm | BBMap | *Bushnell, 2014* | | v.38.18 |
| Software, algorithm | Prinseq | *Schmieder and Edwards, 2011* | | v.0.20.4 |
| Software, algorithm | Kraken2 | *Wood et al., 2019* | | v.2.1.1 |
| Software, algorithm | R | *R Core Team, 2019* | | v.3.6.3–4.1.0 |
| Software, algorithm | Ggplot2 | *Wickham, 2016* | | v.3.0.0 |

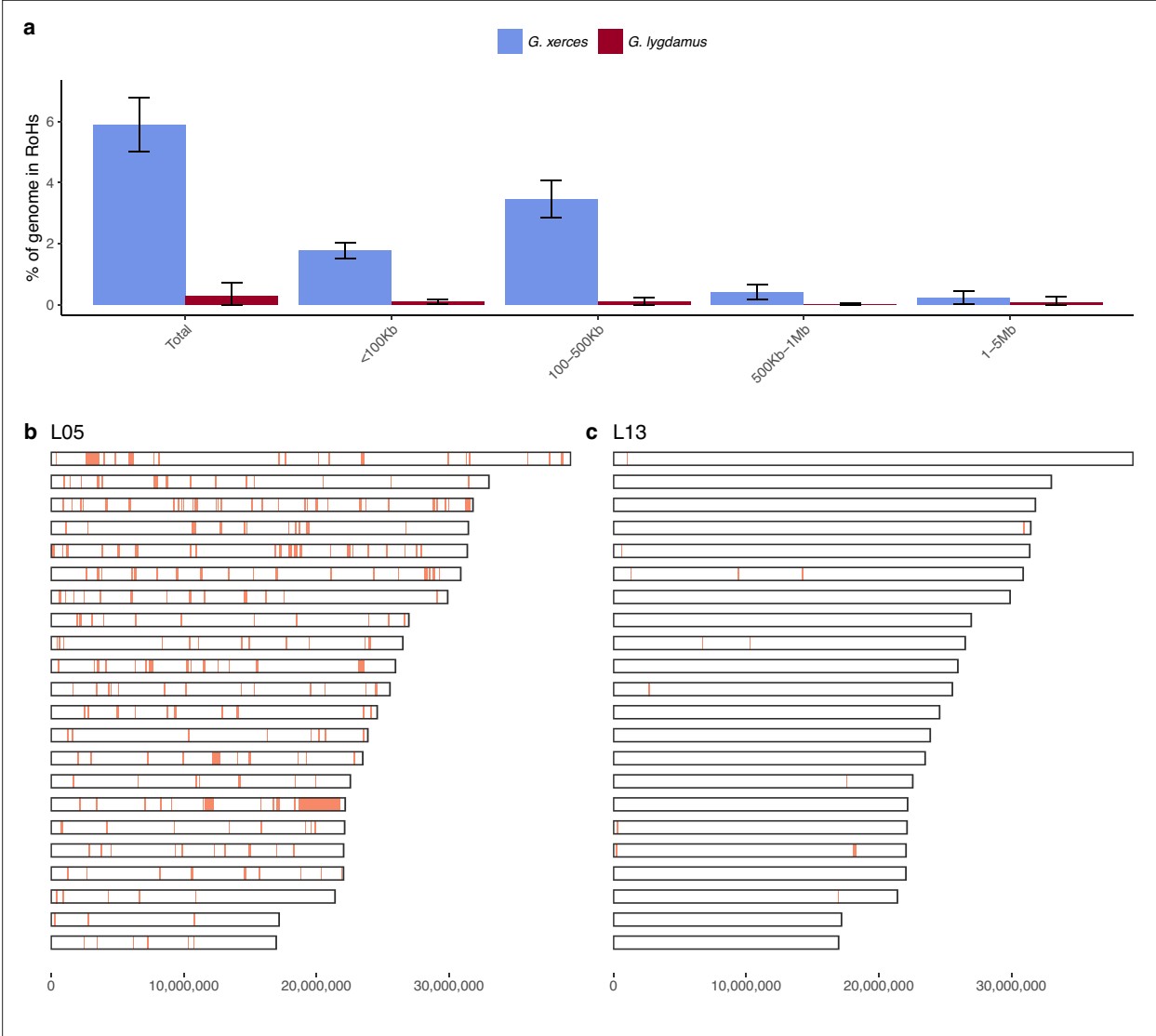

**Figure 4.** Runs of homozygosity (RoH) in the genomes of Xerces Blue and Silvery Blue (modern and historical). (**a**) Percentage of the autosomal genome in RoH by size bins: very short RoH (<100 kb), short RoH (100–500 kb), intermediate RoH (500 kb to 1 Mb), and long (1–5 Mb). Short RoH reflect LD patterns, intermediate size RoH describe background inbreeding due to genetic drift, and long RoH appear in the case of very recent inbreeding due to consanguinity. Error bars show the standard deviation. (**b**) Distribution of RoH in the autosomal genome of a Xerces specimen, L05. (**c**) Distribution of RoH in the autosomal genome of a Silvery specimen L13.

The online version of this article includes the following figure supplement(s) for figure 4:

**Figure supplement 1.** Runs of homozygosity (RoH) in the genomes of Xerces Blue, Silvery Blue, and Green-Underside Blue (modern and historical).

## Historical butterfly specimens

The Xerces Blue specimens analysed belong to the Barnes collection deposited at the Smithsonian National Museum of Natural History. Two of them were collected on 26 April 1923. The Silvery Blue specimens were mostly collected between 1927 and 1948, in Haywood City, Santa Cruz, Oakland, San José, Fairfax, and Marin County (these locations surround San Francisco Bay) (*Table 1*).

## DNA extraction and sequencing of Xerces Blue and Silvery Blue specimens

All DNA extraction and initial library preparation steps (prior to amplification) were performed in a dedicated clean lab, physically isolated from the laboratory used for post-polymerase chain reaction (PCR) analyses. Strict protocols were followed to minimise the amount of human DNA in the ancient

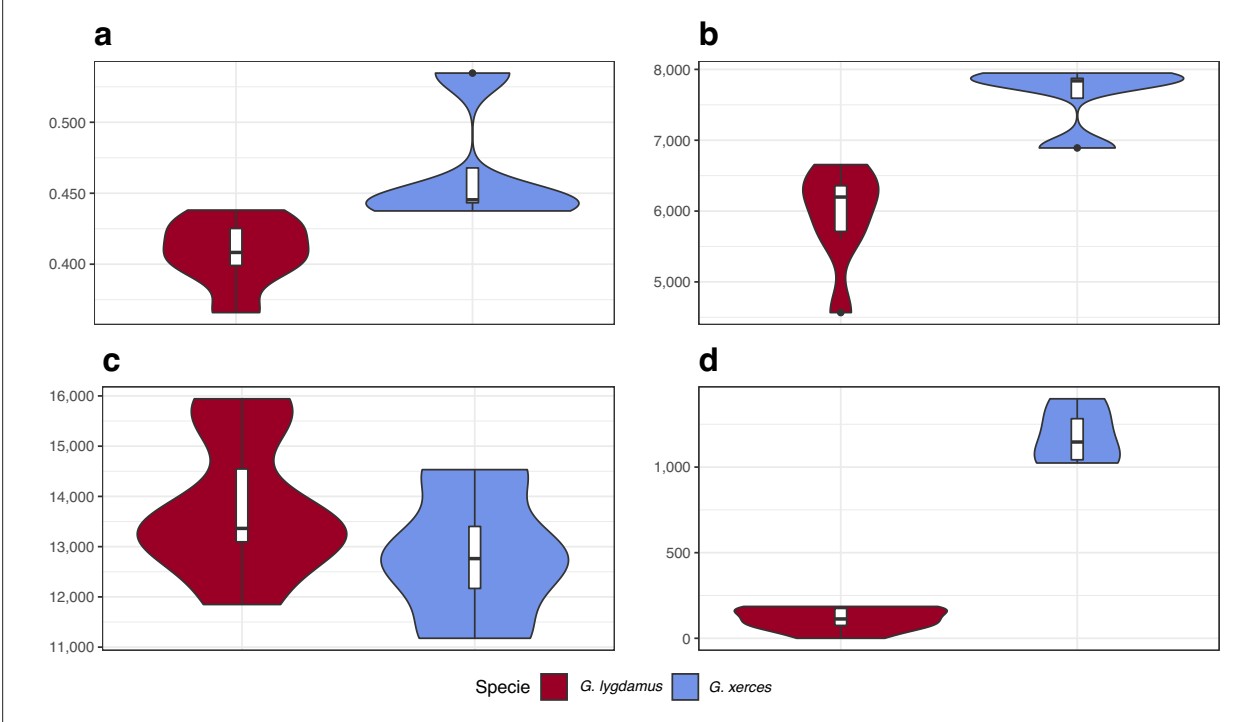

**Figure 5.** Functional effect prediction on the fixed amino acid-changing alleles observed in Xerces Blue and Silvery Blue. (**a**) Wide genome Ka/Ks ratio comparison. (**b**) High-to-moderate effect variant comparison in homozygous sites. (**c**) High-to-moderate effect variant comparison in heterozygous sites. (**d**) Presence of high-to-moderate variants in regions of the genome in runs of homozygosity (RoH). Error bars show the standard deviation.

DNA laboratory, including the wearing a full body suit, sleeves, shoe covers, clean shoes, facemask, hair net, and double gloving, as well as frequent bleach cleaning of benches and instruments. DNA extraction was performed from 12 abdominal samples of historical Xerces Blue and Silvery Blue, as well as a modern Silvery Blue specimen from Canada.

For the extraction procedure, 1 ml of digestion buffer (final concentrations: 3 mM $CaCl_2$, % SDS (sodium dodecyl sulfate), 40 mM DTT (dithiothreitol), 0.25 mg/ml proteinase K, 100 mM Tris buffer pH 8.0, and 100 mM NaCl) was added to each crushed butterfly residue, including an extraction blank, and incubated at 37°C overnight (24 hr) on rotation (750–900 rpm). Next, DNA extraction was continued following the method proposed by *Dabney et al., 2013*. Remaining butterfly sample was then pelleted by centrifugation in a bench-top centrifuge for 2 min at maximum speed ($16,100 \times g$). The supernatant was added to 10 ml of binding buffer (final concentrations: 5 M guanidine hydrochloride, 40% (vol/vol) isopropanol, 0.05% Tween-20, and 90 mM sodium acetate (pH 5.2)) and purified on a High Pure Extender column (Roche). DNA extracts were eluted with 45 µl of low EDTA (Tris-ethylene-diamine-tetraacetic acid) TE buffer (pH 8.0) and quantified using a Qubit instrument.

Following extraction, the DNA extract was converted into Illumina sequencing libraries following the BEST protocol (*Carøe, 2018*). Each library was amplified by PCR using two uniquely barcoded primers, prior to being purified with a 1.5x AMPure clean (Beckman Coulter) and eluted in 25 µl of low EDTA TE buffer (pH 8.0). One Xerces Blue sample did not yield detectable DNA in two independent extractions. For each of the successful extracts, we prepared a single library which was shotgun sequenced on the HiseqX Illumina platform.

### *G. alexis* genome sequencing and annotation

*G. alexis* was chosen as a congeneric reference to compare the demographic histories of both the Xerces Blue and the Silvery Blue. We generated a *G. alexis* reference genome from a male specimen collected in Alcalá de la Selva in Teruel (Spain). Its genome has a sequence length of 619,543,730 bp on 24 chromosomes – including the Z sex chromosome – and the mitochondrial genome. The genome sequence is biologically complete (BUSCO Lepidoptera completeness 97.1%) (*Manni et al., 2021*). The *G. alexis* genome was sequenced at the Sanger Institute as part of the Darwin Tree of Life Project

following the extraction, sequencing, and assembly protocols developed for Lepidoptera (*Hinojosa Galisteo et al., 2021*).

## Xerces Blue and Silvery Blue mapping and variant calling

The ancient DNA reads were clipped using AdapterRemoval2 (*Schubert et al., 2016*), and only reads longer than 25 bp were kept. Filtered reads were mapped against the *G. alexis* assembly with Burrows-Wheeler Aligner (BWA) (*Li and Durbin, 2009*) backtrack algorithm, with parameters optimised for the analysis of aDNA (-l 2, -n 0.01, -o 2). After mapping, duplicated reads were removed using picard MarkDuplicates. Mapped reads with mapping quality below 30 were removed using samtools. Finally, to avoid problems in the next steps derived from spurious callings due to aDNA at reads' ends, we trimmed 2 nt from each read end using BamUtil trimbam. Basic mapping statistics were generated using Qualimap2 (*Okonechnikov et al., 2016*; *Supplementary file 1*). We used bedtools (*Quinlan and Hall, 2010*) to assess genome coverage across the reference using windows of 1 mbp for the nuclear fraction of the genome, as well as depth of coverage, read length, and edit distance distribution. Authenticity of the sequences was assessed by characterising aDNA damage patterns with pmdtools (*Skoglund et al., 2014*) and MapDamage2 (*Jónsson et al., 2013*).

We used snpAD (*Prüfer, 2018*), a program for genotype calling in ancient specimens. The mapped sequences were transformed from bam-format into snpAD-format files, priors for base composition estimated, and genotypes were called using standard settings. The variant call formats (VCFs) were combined and concatenated with CombineVariants and GatherVcfs from GATK (*McKenna et al., 2010*) and filtered with vcftools (*Danecek et al., 2011*) to keep only sites within the mappable fraction of the genome previously obtained with minimum read depth of 2, max read depth of 30, genotype quality >30, maximum missingness of 0.6, minor allele frequency of 5%, and excluding indels and multiallelic sites. Since RVcoll10-B005 is a modern individual, we proceed to map it against the *G. alexis* reference genome with slightly altered parameters. As with the historical samples, pair was collapsed using AdapterRemoval2. BWA mem with default parameters was used as the mapping algorithm. As with the other samples, reads were filtered with Samtools (min. quality of 30) and duplicates were removed by coordinate with picard.

Genotype likelihoods were obtained with ANGSD (*Korneliussen et al., 2014*) using the GATK model with the following parameters for all the samples: -uniqueOnly 1 -remove_bads 1 -only_proper_pairs 1 -trim 10 -C 50 -baq 1 -minInd 5 -skipTriallelic 1 -GL 2 -minMapQ 30.

## Sex determination

The sex of the specimens was determined by differential coverage of the Lepidopteran Z chromosome (females are the heterogametic sex in the Lepidoptera and show reduced coverage on the Z chromosome) (*Supplementary file 1*).

## Mitochondrial phylogenetic tree and divergence dating

Haploid variants were called using bcftools (*Danecek et al., 2021*) with a ploidy of 1, filtering low-quality indels and variants, after which a consensus sequence was exported. We downloaded 14 complete mitochondrial genomes for Polyommatinae from NCBI (*Supplementary file 1*).

All mitochondrial genomes were annotated with MitoFinder (*Allio et al., 2020*) using *Shijimiaeoides divina* as the reference. The 11 protein-coding genes were aligned with the codon-aware aligner MACSE (*Ranwez et al., 2018*) and the ribosomal rRNAs were aligned with MAFFT l-ins-i (*Katoh and Standley, 2013*). We first investigated phylogenetic relationships among five *G. xerces* and eight *G. lygdamus* individuals, with *G. alexis* as the outgroup. We used IQ-TREE2 (*Minh et al., 2020*) to select the best fitting nucleotide substitution model for each partition and merge similar partitions (*Kalyaanamoorthy et al., 2017*), built a maximum likelihood tree and assessed support with 1000 ultrafast bootstrap replicates (*Hoang et al., 2018*).

To infer a time-calibrated phylogenetic hypothesis, we selected one individual of Xerces Blue (L003) and Silvery Blue (RVcoll10-B005) and analysed with 13 other Polyommatinae species. We used BEAST2 (*Ranwez et al., 2018*) with the bModelTest (*Bouckaert and Drummond, 2017*) package to perform phylogenetic site model averaging for each of the merged partitions. Because there is no accepted molecular clock rate for butterflies and no fossils to apply in this part of the phylogeny, we used two strategies to apply time constraints to the analysis. First, we used

two published molecular clock rates for the mitochondrial COX1 gene (1.5% divergence/Ma) estimated for various invertebrates (*Quek et al., 2004*), and the 'standard' insect mitochondrial clock (2.3% divergence/Ma) (*Van Zandt Brower, 1994*). We applied a strict clock with a normal prior set up to span 1.5–2.3% with the 95% HPD interval (mean = 1.9%, sigma = 0.00119). Second, we borrowed the age of the most recent common ancestor of our sampled taxa from fossil-calibrated analyses across butterflies (*Chazot et al., 2019*; *Wiemers et al., 2020*). We fixed the root age to 33 Ma and allowed the remaining node ages to be estimated using a strict clock. Analyses were run twice from different starting seeds for 10 million MCMC generations and trees were sampled every 1000 generations. Runs were checked for convergence with Tracer and all effective sample size values were >200. Runs were combined with the BEAST2 package LogCombiner (*Drummond and Rambaut, 2007*), after removing the first 10% of topologies as burn-in, and a maximum credibility tree was generated with TreeAnnotator (*Drummond and Rambaut, 2007*). Phylogenetic analyses were performed on the National Life Science Supercomputing Center – Computerome 2.0 (https://www.computerome.dk/).

## Xerces Blue and Silvery Blue population histories

We used the PSMC model (*Li and Durbin, 2011*) to explore the demographic history of both butterfly species. We obtained a consensus fastq sequence of the mappable fraction of the genome for each autosomal chromosome (total of 22 chromosomes of *G. alexis* assembly). Only positions with a depth of coverage above 4× and below 15× were kept. Posteriorly, a PSMC was built using the following parameters: -N25 -t15 -r5 -p '28*2+3+5". We used 1 year for the generation time and a mutation rate of $1.9 \times 10^{-9}$, estimated in *Heliconius melpomene* (*Martin et al., 2016*). Considering that calling consensus sequences from low-coverage samples (<10×) can underestimate heterozygous sites (*Keightley et al., 2015*), and given the different coverage between samples, we corrected by false negative rate the samples with coverage lower than the coverage of L005 (for Xerces Blue) and L013 (for Silvery Blue), as recommended by the developers of the software, so that all samples are comparable with each other. However, since in our dataset we do not reach a coverage >20×, we acknowledge that we are not capturing the whole diversity and thus our PSMC might infer lower historical effective population sizes.

## Population stratification and average genome heterozygosity

PCA was performed using PCAngsd (*Meisner and Albrechtsen, 2018*) after obtaining genotype likelihoods with ANGSD including all individuals. To assess global levels of heterozygosity, the unfolded site frequency spectrum (SFS) was calculated for each sample separately using ANGSD (*Korneliussen et al., 2014*) and realSFS with the following quality filter parameters: -uniqueOnly 1 - remove_bads 1 -only_proper_pairs 1 -trim 10 -C 50 -baq 1 -minMapQ 30 -minQ 30 -setMaxDepth 200 - doCounts 1 -GL 2 -doSaf 1.

## Runs of homozygosity

RoH were called based on the density of heterozygous sites in the genome using the implemented hidden Markov model in bcftools (*Danecek et al., 2021*) roh with the following parameters: `-G30 --skip-indels --AF-dflt 0.4 --rec-rate` $1e^{-9}$ from the mappable fraction of the genome with the filtered VCF file. We kept the RoH with a phred score >85. We divided the RoH into different size bins: very short RoH (<100 kb), short RoH (100–500 kb), intermediate RoH (500 kb to 1 Mb), and long (1–5 or >5 Mb). Short RoH reflect LD patterns, intermediate size RoH describe background inbreeding due to genetic drift, and long RoH appear in the case of recent inbreeding (*Ceballos et al., 2018*).

## Deleterious load

We used the *G. alexis* annotations to create a SNPeff database that we used to annotate our callings. Using SNPeff (*Cingolani et al., 2012*) again and the set of variants discovered by angsd, we predicted the putative effect of those variant in the analysed individuals (*Supplementary file 1*). In addition to wide genome mutations, we specifically focussed on mutations present in homozygosis, heterozygosis, and the previously annotated RoH.

**Table 4.** *Wolbachia* DNA reads assigned using Kraken2.

| Specimen | *Wolbachia* genus reads | *Wolbachia* spp. reads |
|---|---|---|
| L002 | 190 | 5 |
| L003 | 131 | 3 |
| L004 | 213 | 5 |
| L005 | 311 | 8 |
| L006 | 242 | 9 |
| L007 | 152 | 2 |
| L008 | 414 | 21 |
| L009 | 236 | 6 |
| L011 | 184 | 9 |
| L012 | 168 | 9 |
| L013 | 523 | 24 |

## Unrecoverable regions

To further explore how the genomic divergence can influence our genome reconstruction success, we undertook a similar approach as the genome of the Christmas Island rat (*Lin et al., 2022*), and explored the chromosomal regions in the *G. alexis* reference that were significantly depleted of Xerces DNA reads. We used bedtools (*Quinlan and Hall, 2010*) and some in-home bash scripting to calculate the mean coverage per gene of the *G. alexis* genome for Xerces Blue sequencing DNA reads. We first used bedtools' algorithms *bamtobed* and *genomecov* to estimate the genome-wide per-site coverage of the reference genome in these two species. Then, we extracted the coordinates of all protein-coding genes from the annotation file (gff file) and used the intersect to estimate the average coverage of each protein-coding gene. We performed a functional analysis of all genes uncovered in *G. alexis*, excluding those that are present in *G. lygdamus* with more than 5× coverage (as we were looking for evolutionary novelties in the Xerces Blue lineage alone) using profile- IntersProScan (*Jones et al., 2014*) and sequence similarity-based (blasp) searches (*Gish and States, 1993*; *Supplementary file 1*).

## Colouration genes variability

To find possible amino acid-changing variants that could explain phenotypical differences between *G. lygdamus* and *G. xerces*, we have identified and explored three well-known genes associated to colour patterns in butterflies: *optix*, *cortex*, and *Wnt* genes (*Zhang and Reed, 2016*; *Zhang et al., 2017*; *Mazo-Vargas et al., 2017*; *Fenner et al., 2020*; *Banerjee et al., 2021*). First, we located those genes in our annotation with BLAST and their homologs in other butterfly species, setting an *E*-value lower than 0.001 and an Identity value above 60% (*Table 3*). Then, the coordinates were called using GATK UnifiedGenotyper. Variants were filtered for indels and minimum Genotype Quality of 30 using. Variants were kept regardless of their coverage. A variant is considered as fixed in one species if it is covered in at least two individuals of each species, it is in homozygous state, and when one of the species present all their genotypes calls as homozygous for the alternative allele while in the other are homozygous of the reference allele. No fixated mutations were identified in the regions covered at the same time by *G. lygdamus* and *G. xerces* sequences.

## *Wolbachia* screening

*Wolbachia* are endosymbiotic alpha-proteobacteria that are present in about 70% of butterfly species and induce diverse reproductive alterations, including genetic barriers when two different strains infect the same population or when two populations – one infected and one uninfected – meet (*Telschow et al., 2005*). As potential evidence for a reproductive barrier promoting the separation of Xerces Blue and Silvery Blue, we searched for *Wolbachia* DNA reads in our specimens, taking advantage of the high coverage and the shotgun approach. First, we collapsed unique reads from the butterfly-free sequences with BBmap (*Bushnell, 2014*) and removed from the dataset low complexity sequences using Prinseq (*Schmieder and Edwards, 2011*). Afterwards, we used

Kraken2 (*Wood et al., 2019*) to assign reads against the standard plus human Kraken2 database (bacteria, archaea, fungi, protozoa, and viral). The historical specimens did not display enough reads assigned to *Wolbachia* for us to suspect of the presence of the bacteria in those samples (*Table 4*).

## Acknowledgements

CL-F is supported by a PID2021-124590NB-100 grant (MCIU/AEI/FEDER, UE) of Spain; TM-B is supported by funding from the European Research Council (ERC) (grant agreement No. 864203), BFU2017-86471-P (MINECO/FEDER, UE), 'Unidad de Excelencia María de Maeztu', funded by the AEI (CEX2018-000792-M), Howard Hughes International Early Career, and Generalitat de Catalunya, GRC 2017-SGR-880; RV is supported by grant PID2019-107078GB-I00, funded by MCIN/AEI/10.13039/501100011033, and by GRC 2017-SGR-991 (Generalitat de Catalunya). We are grateful to the SCIENCE Faculty at University of Copenhagen for free access to Computerome 2.0. This research was funded in whole, or in part, by the Wellcome Trust Grants 206194 and 218328 (MU, MB). For the purpose of Open Access, the author has applied a CC BY public copyright license to any Author Accepted Manuscript version arising from this submission.

## Additional information

### Funding

| Funder | Grant reference number | Author |
|---|---|---|
| Ministerio de Ciencia e Innovación | PID2021-124590NB-100 | Carles Lalueza-Fox |
| European Research Council | 864203 | Tomàs Marquès |
| Ministerio de Ciencia e Innovación | BFU2017-86471-P | Tomàs Marquès |
| Ministerio de Ciencia e Innovación | CEX2018-000792-M | Tomàs Marquès |
| Generalitat de Catalunya | GRC 2017-SGR-880 | Tomàs Marquès |
| Wellcome Trust | 10.35802/206194 | Marcela Uliano-Silva |
| Wellcome Trust | 10.35802/218328 | Mark Blaxter |
| Ministerio de Ciencia e Innovación | PID2019-107078GB-I00 | Roger Vila |
| Generalitat de Catalunya | GRC 2017-SGR-991 | Roger Vila |
| Howard Hughes Medical Institute | Howard Hughes International Early Career | Tomàs Marquès |

The funders had no role in study design, data collection, and interpretation, or the decision to submit the work for publication. For the purpose of Open Access, the authors have applied a CC BY public copyright license to any Author Accepted Manuscript version arising from this submission.

### Author contributions

Toni de-Dios, Charlotte Wright, Formal analysis, Investigation; Claudia Fontsere, Formal analysis, Methodology; Pere Renom, Conceptualization, Investigation; Josefin Stiller, Formal analysis, Investigation, Visualization, Writing – original draft; Laia Llovera, Resources, Formal analysis; Marcela Uliano-Silva, Esther Lizano, Conceptualization, Formal analysis; Alejandro Sánchez-Gracia, Methodology; Berta Caballero, Sergi Civit, Investigation; Arcadi Navarro, Resources, Supervision, Project administration; Robert K Robbins, Conceptualization, Writing – original draft, Writing – review and editing; Mark Blaxter, Conceptualization, Supervision; Tomàs Marquès, Resources, Supervision, Investigation, Project administration; Roger Vila, Conceptualization, Supervision, Writing – original draft; Carles

Lalueza-Fox, Conceptualization, Funding acquisition, Writing – original draft, Project administration, Writing – review and editing

### Author ORCIDs
Toni de-Dios ⓘ https://orcid.org/0000-0001-9260-8846
Roger Vila ⓘ https://orcid.org/0000-0002-2447-4388
Carles Lalueza-Fox ⓘ https://orcid.org/0000-0002-1730-5914

Reviewer #1 (Public Review): https://doi.org/10.7554/eLife.87928.3.sa1
Reviewer #2 (Public Review): https://doi.org/10.7554/eLife.87928.3.sa2
Author response https://doi.org/10.7554/eLife.87928.3.sa3

## Additional files

### Supplementary files
• Supplementary file 1. Additional information on the genomics analyses. (a) DNA metric. (b) Mappability of the reference genomes. (c) Mitochondrial DNA genomes used in the phylogenetic analysis. (d) Heterozygosity, homozygosity, and runs of homozygosity (RoH). (e) Genes in uncovered genomic regions in Xerces Blue.

• MDAR checklist

### Data availability
The accession numbers for the Xerces Blue and Silvery Blue genomes reported in this study are in the European Nucleotide Archive (ENA): PRJEB47122. Data on *G. alexis* are available in INSDC under BioProject PRJEB43798 and genome assembly accessions GCA_905404095.1 (primary haplotype) and GCA_905404225.1 (secondary, alternate haplotype).

The following datasets were generated:

| Author(s) | Year | Dataset title | Dataset URL | Database and Identifier |
|---|---|---|---|---|
| de-Dios T, Fontsere C, Renom P, Stiller J, Llovera L, Uliano-Silva M, Sánchez-Gracia A, Wright C, Lizano E, Caballero B, Navarro A, Civit S, Robbins RK, Blaxter M, Marquès-Bonet T, Vila R, Lalueza-Fox C | 2023 | Whole-genomes from the extinct Xerces Blue butterfly reveal low diversity and long-term population decline | https://www.ebi.ac.uk/ena/browser/view/PRJEB47122 | European Nucleotide Archive, PRJEB47122 |
| de-Dios T, Fontsere C, Renom P, Stiller J, Llovera L, Uliano-Silva M, Sánchez-Gracia A, Wright C, Lizano E, Caballero B, Navarro A, Civit S, Robbins RK, Blaxter M, Marquès-Bonet T, Vila R, Lalueza-Fox C | 2021 | ilGlaAlex (green-underside blue) | https://www.ncbi.nlm.nih.gov/bioproject/?term=PRJEB43798 | NCBI BioProject, PRJEB43798 |
| de-Dios T, Fontsere C, Renom P, Stiller J, Llovera L, Uliano-Silva M, Sánchez-Gracia A, Wright C, Lizano E, Caballero B, Navarro A, Civit S, Robbins RK, Blaxter M, Marquès-Bonet T, Vila R, Lalueza-Fox C | 2021 | Genome assembly ilGlaAlex1.1 | https://www.ncbi.nlm.nih.gov/datasets/genome/GCA_905404095.1/ | NCBI GenBank, GCA_905404095.1 |

*Continued on next page*

*Continued*

| Author(s) | Year | Dataset title | Dataset URL | Database and Identifier |
|---|---|---|---|---|
| de-Dios T, Fontsere C, Renom P, Stiller J, Llovera L, Uliano-Silva M, Sánchez-Gracia A, Wright C, Lizano E, Caballero B, Navarro A, Civit S, Robbins RK, Blaxter M, Marquès-Bonet T, Vila R, Lalueza-Fox C | 2021 | Genome assembly ilGlaAlex1.1 alternate haplotype | https://www.ncbi.nlm.nih.gov/datasets/genome/GCA_905404225.1/ | NCBI GenBank, GCA_905404225.1 |

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
