## [Editor Report · eLife assessment]

This **important** study illustrates the value of museum samples for understanding past genetic variability in the genomes of populations and species, including those that no longer exist. The authors present genomic sequencing data for the extinct Xerces Blue butterfly and report **convincing** evidence of declining population sizes and increases in inbreeding beginning 75,000 years ago, which strongly contrasts to the patterns observed in similar data from its closest relative, the extant Silvery Blue butterfly. Such long-term population health indicators may be used to highlight still extant but especially vulnerable-to-extinction insect species – irrespective of their current census population size abundance.

---

## [Referee Report · Reviewer #1 (Public Review)]

The authors report a study, where they have sequenced whole genomes of four individuals of an extinct species of butterfly from western North America (Glaucopsyche xerces), along with seven genomes of a closely related species (Glaucopsyche lygdamus), mainly from museum specimens, several to many decades old. They then compare these fragmented genomes to a high-quality, chromosome-level assembly of a genome of a European species in the same genus (Glaucopsyche alexis). They find that the extinct species shows clear signs of declining population sizes since the last glacial period and an increase in inbreeding, perhaps exacerbating the low viability of the populations and contributing to the extinction of the species.

The study really highlights how museum specimens can be used to understand the genetic variability of populations and species in the past, up to a century or more ago. This is an incredibly valuable tool, and can potentially help us to quickly identify whether current populations of rare and declining species are in danger due to inbreeding, or whether at least their genetic integrity is in good condition and other factors need to be prioritised in their conservation. In the case of extinct species, sequencing museum specimens is really our only window into the dynamics of genomic variability prior to extinction, and such information can help us understand how genetic variation is related to extinction.

I think the authors have achieved their goal admirably, they have used a careful approach to mapping their genomic reads to a related species with a high-quality genome assembly. They might miss out on some interesting genetic information in the unmapped reads, but by and large, they have captured the essential information on genetic variability within their mapped reads. Their conclusions on the lower genetic variability in the extinct species are sound, and they convincingly show that Glaucopyche xerces is a separate species to Glaucopsyche lygdamus (this has been debated in the past).

---

## [Referee Report · Reviewer #2 (Public Review)]

The Xerces Blue is an iconic species, now extinct, that is a symbol for invertebrate conservation. Using genomic sequencing of century-old specimens of the Xerces Blue and its closest living relatives, the authors hypothesize about possible genetic indicators of the species' demise. Although the limited range and habitat destruction are the most likely culprits, it is possible that some natural reasons have been brewing to bring this species closer to extinction.

The importance of this study is in its generality and applicability to any other invertebrate species. The authors find that low effective population size, high inbreeding (for tens of thousands of years), and higher fraction of deleterious alleles characterize the Xerces colonies prior to extinction. These signatures can be captured from comparative genomic analysis of any target species to evaluate its population health.

It should be noted that it remains unclear if these genomic signatures are indeed predictive of extinction, or populations can bounce back given certain conditions and increase their genetic diversity somehow.

Methods are detailed and explained well, and the study could be replicated. I think this is a solid piece of work. Interested researchers can apply these methods to their chosen species and eventually, we will assemble datasets to study extinction process in many species to learn some general rules.

---

## [Author Response]

The following is the authors’ response to the original reviews.

**Reviewer #1 (Public Review):**
The authors report a study, where they have sequenced whole genomes of four individuals of an extinct species of butterfly from western North America (Glaucopsyche xerces), along with seven genomes of a closely related species (Glaucopsyche lygdamus), mainly from museum specimens, several to many decades old. They then compare these fragmented genomes to a high-quality, chromosome-level assembly of a genome of a European species in the same genus (Glaucopsyche alexis). They find that the extinct species shows clear signs of declining population sizes since the last glacial period and an increase in inbreeding, perhaps exacerbating the low viability of the populations and contributing to the extinction of the species.The study really highlights how museum specimens can be used to understand the genetic variability of populations and species in the past, up to a century or more ago. This is an incredibly valuable tool, and can potentially help us to quickly identify whether current populations of rare and declining species are in danger due to inbreeding, or whether at least their genetic integrity is in good condition and other factors need to be prioritised in their conservation. In the case of extinct species, sequencing museum specimens is really our only window into the dynamics of genomic variability prior to extinction, and such information can help us understand how genetic variation is related to extinction.I think the authors have achieved their goal admirably, they have used a careful approach to mapping their genomic reads to a related species with a high-quality genome assembly. They might miss out on some interesting genetic information in the unmapped reads, but by and large, they have captured the essential information on genetic variability within their mapped reads. Their conclusions on the lower genetic variability in the extinct species are sound, and they convincingly show that Glaucopyche xerces is a separate species to Glaucopsyche lygdamus (this has been debated in the past).

We thank the reviewer for his/her positive assessment and we hope to have contributed to both the knowledge of this iconic extinct species and also the possibility of applying our observations to other, endangered insects.

**Reviewer #2 (Public Review):**
The Xerces Blue is an iconic species, now extinct, that is a symbol for invertebrate conservation. Using genomic sequencing of century-old specimens of the Xerces Blue and its closest living relatives, the authors hypothesize about possible genetic indicators of the species' demise. Although the limited range and habitat destruction are the most likely culprits, it is possible that some natural reasons have been brewing to bring this species closer to extinction.The importance of this study is in its generality and applicability to any other invertebrate species. The authors find that low effective population size, high inbreeding (for tens of thousands of years), and higher fraction of deleterious alleles characterize the Xerces colonies prior to extinction. These signatures can be captured from comparative genomic analysis of any target species to evaluate its population health.It should be noted that it remains unclear if these genomic signatures are indeed predictive of extinction, or populations can bounce back given certain conditions and increase their genetic diversity somehow.Methods are detailed and explained well, and the study could be replicated. I think this is a solid piece of work. Interested researchers can apply these methods to their chosen species and eventually, we will assemble datasets to study extinction process in many species to learn some general rules.

We thank the reviewer for his/her observations and suggestions for improvement and we agree that endangered species show conflicting signals sometimes associated to decreasing genetic diversity (some species are very low in numbers and yet they keep reasonably high diversity levels as compare to others); however, this aspect remains to be explored in detail in insects that have demographic dynamics to a large extent impossible to compare to those observed in vertebrates. We agree there is a full range of cases and circumstances in declining insects to be explored in the future.

Several small questions/suggestions:1. The authors reference a study concluding that Shĳimiaeoides is Glaucopsyche. Their tree shows the same, confirming previous publications. And yet they still use Shĳimiaeoides, which is confusing. Why not use Glaucopsyche for all these blues?

We have decided, for the sake of clarity, to change it to Glaucopsyche divina in Figure 1, as suggested by the reviewer.

1. Plebejus argus is a species much more distant from P. melissa than Plebejus anna (anna and melissa are really very close to each other), and yet their tree shows the opposite. What is the problem? Misidentification? Errors in phylogenetic analyses?

The reviewer is right and we think there is a mixture of potential problems here that deserve a more in depth analysis of this genus. We used MN974526 as a proxy for P. argus and we suspect now this is probably a case of misidentification (but we cannot verify it without a morphological examination of the original specimen and likely additional genomic data). MN974526 shows a 99.33% identity to the sequence by Vila et al. (2011) code NGK02C411, defined as P. melissa; as the true status of this mitogenome cannot be totally clarified (it is likely that it is in fact P. idas), we have decided to attribute it to “Plebejus sp” in the Figure 1 and explained this in the text.

1. Wouldn't it be nicer to show the underside of butterfly pictures that reveals the differences between xerces and others? Now, they all look blue and like one species, no real difference.

This is a good suggestion, and we have now included the underside of different species, including Xerces Blue.

1. The authors stated that one of five xerces specimens failed to sequence, and yet they show 5 specimens in the tree. Was the extra specimen taken from GenBank?

Yes, the extra specimen is the one reported in Grewe et al. 2021; we have marked in Figure 1 with an * this specific mitogenome (and mentioned in the legend), which clusters nicely within the set of Xerces Blue mtDNA diversity we have generated.

**Reviewer #1 (Recommendations For The Authors):**
I am curious why the authors did not attempt to do a de novo assembly of the extinct species' genomes. In our work on museum specimen genomes, we have successfully used a de novo approach to extract protein coding genes from such highly fragmented genomes. We used SPAdes to assemble the museum genomes and then assessed BUSCO completeness, finding anything from 50% to 90% BUSCO completeness. The genome assemblies themselves are pretty poor with N50s around a few thousand bp at best, but the information we can extract from such highly fragmented genomes is very useful, especially with regard to protein coding gene exons. Perhaps worth trying?

Thanks for the comment. In our approach, and considering the expected low quality from some museum specimens in the lower part of the conservation spectrum, we used the standard approach based on the variant calling of short read data mapped to a close assembly. This method has been shown to be precise enough in cross species mapping (Kuderna et al. Science 2023). Local assemblies of exons and genes, while potentially informative, particularly for structural preservation, was not the priority in our objectives where only the base pair mutations were explored. Nevertheless, we are planning to generate in the near future an assembly for the closest living relative of Xerces, Glaucopsyche lygdamus, and once we get it, we will consider the possibility of undertaking the suggested approach with this new reference to explore the genomic architecture of Xerces Blue in more detail.